# Enrichment and Chemical Speciation of Vanadium and Cobalt in Stone Coal Combustion Products in Ankang, Shanxi Province, China

**DOI:** 10.3390/ijerph19159161

**Published:** 2022-07-27

**Authors:** Wei Cui, Qingjun Meng, Wenbo Li, Qiyan Feng

**Affiliations:** 1School of Environmental Science and Spatial Informatics, China University of Mining and Technology, Xuzhou 221116, China; cuiweikj@163.com (W.C.); liwb1218@163.com (W.L.); fqycumt@126.com (Q.F.); 2Shaanxi Huacheng Industrial Joint Stock Company, Xi’an 710000, China

**Keywords:** stone coal, vanadium, cobalt, enrichment, combustion products, chemical speciation

## Abstract

Stone coal enriches more elements compared to other coals, especially Vanadium (V). The content of Co is relatively low, while its environmental risk is relatively high. This study collected the stone-coal samples to investigate the enrichment characteristics and the chemical speciation in the combustion products of V and Co in stone coal at an open-pit coal mine in Ankang City, Shanxi Province, China. A simulation combustion experiment and sequential chemical extraction were conducted. Mineral composition was analyzed for raw-stone coal and its combustion products. The results showed that most of V and Co are mainly enriched in combustion products during the combustion process, the enrichment capacity of Co is higher than V. With the increase in the combustion temperature, the bioavailable chemical speciation of V in stone coal combustion products increased, while Co decreased. If the combustion products are stored without effective treatment, the surrounding environment will be polluted, and then human health might be endangered.

## 1. Introduction

Stone coal is a kind of low quality sapropelic coal formed in the early Proterozoic and early Paleozoic [1]. It has the characteristics of high ash, high sulfur, a low calorific value and many associated elements [2]. It is not only a type of low carbon and low quality anthracite with a low calorific value, but it is also a low grade metal symbiotic ore, with vanadium as the main component and multiple metal elements [3,4,5]. High-quality stone coal with a high carbon content is usually black, with half-bright luster and less impurities; the stone coal, with a lower carbon content, is grayish, dark and lightless, and contains more pyrite, quartz veins and phosphorus calcium nodules [6].

The southern Shaanxi Province of China is famous, at home and abroad, for the occurrence of stone coal resources [7,8,9]. The main stone coal-bearing strata are the Lower Cambrian Lujiaping Formation [5]. The elements V, Mo, Ga, P, Cr, Pb, Ni and others are mainly associated with the stone coal in southern Shaanxi [10,11]. Although stone coal has the disadvantages of a low calorific value and high harmful elements, it has a long history of mining and utilization in areas with scarce resources. At present, the main utilization methods of stone coal resources in China are stone coal power generation, stone coal vanadium extraction and building materials production [12,13]. The raw materials of coal, with different geological ages and different metamorphic degrees, are different, and the internal composition, structure and properties of coal also show corresponding changes, which lead to different characteristics of coal oxidation and combustion processes [14]. Current researches on the combustion characteristics of stone coal usually use coal samples of different geological ages as fuel [15], to study the combustion kinetics [16] or to find ways and means to purify flue gas and ash from burning process for the purpose of utilization [17,18].

Vanadium and cobalt play important roles in many fields. Ferrovanadium and metallic vanadium are mainly used in the steel and metallurgy and aerospace industries, while vanadium-containing compounds are used in the chemical and battery industries. It is called “gourmet powder for modern industry” due to its high melting point and good catalytic performance [2]. Co, a silvery-white ferromagnetic metal, is an important raw material for producing heat-resistant alloys, hard alloys, corrosion-resistant alloys, magnetic alloys and various cobalt salts. It is widely used in lithium batteries, catalysts, ceramic pigments, medicines, and the aerospace and other industries. It has a good reputation as “industrial monosodium glutamate” and “ industrial teeth” [19]. Therefore, V and Co in stone coal has received widespread attention. However, V has been proved to be a potentially dangerous chemical pollutant that can harm plants, crops and even the entire agricultural system [20]. V itself is non-toxic, but V compounds are mostly toxic, such as V_2_O_5_, NH_4_VO_3_, etc. V can irritate the eyes, nose, throat, respiratory tract and cause coughs. V is also a poison that can be absorbed by the whole body and can affect the gastrointestinal tract, nervous system and heart. Severe vasospasm occurred in the kidneys and intestinal tract during vanadium poisoning [21]. V in the environment can enter the human body through breathing, drinking water, food and other ways; an intake of a high concentration of V will cause harm to human health. In the list of carcinogens published by the World Health Organization in 2017, Co and Co compounds were listed in the list of 2B carcinogens. The Environmental Quality Standards for Surface Water (GB 3838-2002) stipulates that the concentration of Co in the water should not be greater than 1.0 milligram/L. Co is an essential trace element for plant growth and is a component of vitamin B12. A proper amount of low-concentration Co can promote plant growth, but if the concentration is too high, the plants will be poisoned. If the environment is polluted by radioactive Co, its radioactivity is persistent. As the decay gradually decreases, the radioactivity will seriously affect the growth and development of the surrounding plants and animals. If food containing radioactive Co is eaten, it will lead to alopecia, seriously damage the cellular tissues in human blood, cause leukopenia and cause diseases of blood system [22,23].

Most of the research focuses on the occurrence state of V in stone coal and the extraction process of V [24,25], but little research has been completed on the enrichment of V and Co in stone coal combustion products and the environmental risks of its chemical speciation. As rare metals, V and Co are important strategic resources as well as potentially toxic elements; therefore, it is necessary to conduct basic research on the enrichment, chemical morphology and morphology-related environmental risks in the raw stone coal and its combustion products at different combustion temperatures, which is helpful to understand the industrial value of the stone coal combustion products and the possible environmental risks.

## 2. Materials and Methods

### 2.1. Sample Collection and Preparation

The experimental raw stone coal samples were collected from an open-pit stone coal mine, Haoping river basin, Ankang City, Shaanxi Province, China. Considering the representativeness of the samples in the studied coal mine, three points of the same stone coal seam were selected, two samples were collected at each point, and each sample was about 5 kg. The samples’ collection was completed strictly in accordance with GB475-2008 [26]. The collected coal samples were sealed and carried in polyethylene sample bags to prevent pollution, weathering and loss during transportation. All of the stone coal samples were uniformly mixed in the laboratory for the experiment. After being brought back to the laboratory, the stone coal samples were air-dried and crushed to pass a 200-mesh sieve for analysis.

### 2.2. Proximate Analysis

A proximate analysis (moisture, ash yield, volatile matter and fixed carbon) was performed on the raw stone coal sample, according to the Chinese coal standard GB/T 212-2008, which is comparable to ASTM standards [27] (ASTM-D3173-11, 2011; ASTM-D3174-11, 2011; ASTM-D3175-11, 2011), except for a small temperature difference used in the measurement procedures. According to the Chinese standards GB/T 214-2007 (equivalent to ASTM standard D3177-02, 2002) and GB/T 215-2003 [28] (equivalent to ASTM standard D2492-02, 2012), a WS-S101 automatic sulfur analyzer was used to measure the total sulfur and various forms of sulfur, respectively.

### 2.3. Simulated Combustion Experiment

Six combustion temperature points were selected between 500 °C and 1000 °C at 100 °C intervals to study the release behavior of the elements, V and Co, during the stone coal combustion. 2 g of raw stone coal sample was placed in a crucible, covered and placed in a muffle furnace, where the temperature was increased to the desired points followed by 4 h dwelling. Then, the heated coal sample was taken out from the muffle roaster when the temperature dropped to room temperature, and stored in a dryer for later use. We used the following notations to name the raw and the combustion products: RC (experimental raw stone coal); P500 (stone coal combusted at 500 °C); P600 (stone coal combusted at 600 °C); P700 (stone coal combusted at 700 °C); P800 (stone coal combusted at 800 °C); P900 (stone coal combusted at 900 °C); P1000 (stone coal combusted at 1000 °C), respectively.

### 2.4. Sequential Chemical Extraction Procedure

The chemical speciation is a crucial factor in determining the environmental fates of the heavy metals in coal during combustion, weathering, conversion, leaching and cleaning [29]. Both direct (e.g., X-ray absorption fine structure, electron microprobe analysis) and indirect methods (e.g., mathematical statistics, float and sink technique and sequential chemical extraction) were used to study the occurrence of the heavy elements in coals. Among them, sequential chemical extraction (SCE) is especially effective as a quantitative analysis method [30]. According to the characteristics of “slow weight loss–fast weight loss–nearly constant weight” in the process of heating combustion, RC, P600 and P1000 were selected to represent the above three stages, respectively. The method of Tessier [31] was adopted to carry out the continuous chemical extraction experiments on the above samples, so as to determine the V and Co concentrations in the five solutions obtained in the continuous extraction, and represent the five fractions in the stone coal and its combustion products. In order to ensure the repeatability of the experiment, each sample was treated three times. The sequential chemical extraction is a method that can quantitatively study the speciation of trace elements, which can quantitatively analyze the content of V and Co in five forms and they include exchangeable, carbonate, Fe-Mn oxides, organic and residual bound states, respectively. This method consists of the following five steps:

A total of 1 g of stone coal and combustion product samples were added to 50 mL plastic centrifuge tubes, and 8 mL of 1 mol/L magnesium chloride MgCl_2_·6H_2_O were added to the tubes. Then, this was shaken at room temperature (200 r/min), centrifuged for 10 min (4000 r/min), the supernatant was removed, and the removed solution was filtered for analysis of exchangeable state;

The residue of the previous step was extracted with 8 mL, 1 mol/L sodium acetate solution NaOAc, at room temperature. Before extraction, the pH was adjusted to 5.0 with acetic acid HOA_C_, shaken for 8 h (200 r/min), centrifuged for 10 min (4000 r/min), the supernatant was removed, and the removed solution was filtered to analyze the carbonate binding state;

A further 20 mL 0.04 mol/L NH_2_OH·HCl 25% (*v*/*v*) HOA_C_ solution was added to the residue after the previous step for extraction, the extraction temperature was 96 ± 3 °C, the time was 4 h, centrifugation was 10 min (4000 r/min), the supernatant was removed, and the removed solution was filtered to analyze the oxidation state of iron and manganese.

To the residue treated in the previous step, 3 mL of 0.02 mol/L nitric acid solution HNO_3_ and 5 mL of 30% (*v*/*v*) hydrogen peroxide H_2_O_2_ were added, then the pH was adjusted to 2 with HNO_3_, the mixture was heated to 85 ± 2 °C, kept warm for 2 h, and oscillated several times during heating. Another 5 mL of H_2_O_2_ was added, the pH was adjusted to 2, then the mixture was heated to 85 ± 2 °C, the temperature stabilized for 3 h, and oscillated intermittently. After cooling, 5 mL of 3.2 mol/L ammonium acetate solution NH_4_OAc was added, diluted to 20 mL with 20% (*v*/*v*) HNO_3_, and shaken for 30 min. The mixture was centrifuged for 10 min (4000 r/min), the supernatant remove, and the removed solution was filtered to analyze the organic binding state;

The residue after extracting the organic state was digested with 8 mL of HNO_3_, 2 mL of H_2_O_2_, 3 mL of HF and 4 mL of HClO_4_ in Teflon digestion vessels. Then, the samples were gradually heated on a hot plate from 100 to 210 °C for 12 h. After cooling down, the digested solutions were diluted to 25 mL with 1% of HNO_3_.

### 2.5. Experimental Analysis

The total concentrations of raw stone coal were determined by X-ray Fluorescence spectrometry(XRF, Munich, Germany, Bruker S8 TIGER). All of the extracting solutions of each speciation were determined by Inductively Coupled Plasma Mass Spectroscopy (ICP-MS, Thermo Fisher Scientific, Waltham, MA, USA). The accuracy of the elements was evaluated, using the standard reference material SARM20 (coal) and GBW07406 (GSS-6) (soil). The instrument was calibrated using multi-elemental standard solutions containing known concentrations of V and Co. The calibration curves for all of these elements were linear (R^2^ > 0.996, *n* = 6) over the range of the experimental samples. The recoveries of V and Co are 115.43% and 110.43%, respectively. About 0.1 g of raw stone coal (combusted 3.5 h at 450 °C to remove some of the carbon in the coal and increase the digestion efficiency), and the heated stone coal samples were digested, using the same method as the last step in the chemical continuous extraction by HNO_3_-HF-HClO_4_, then the total concentrations of V and Co were determined by Inductively Coupled Plasma Mass Spectroscopy (ICP-MS, Thermo Fisher Scientific). A Philips X’ Pert PRO X-ray diffractometer (XRD) with Cu K-alpha radiation was used to determine the mineral compositions of RC, P500, P600, P700, P800, P900 and P1000. The XRD pattern was recorded over a 2θ interval of 10–70° with a step increment of 0.01°. The minerals are demarcated in terms of the International Centre for Diffraction Data (ICDD) powder diffraction file. The morphology of the raw coal and its combustion products were observed using a Scanning Electron Microscope (SEM, FEI, P.O., USA).

## 3. Results and Discussion

### 3.1. Proximate Analysis and Chemical Composition of Raw Stone Coal

The proximate analysis results are shown in Table 1. The calorific value of the raw stone coal is 5.33 Mega Joule/Kilogram (MJ/kg), which is far lower than that of standard coal at 29.29 MJ/kg. The percentage of total sulfur in the raw coal is 1.82%, of which sulfide accounts for more than 96%. Combined with the XRD analysis, it can be seen that the main sulfur form is pyrite sulfur. The result shows that the raw stone coal has a high ash content, low moisture, volatile substances and calorific value and a medium sulfur content. The chemical analysis of the raw stone coal is shown in Table 2. As can be seen from the composition, the carbon content of the stone coal in the study area is high, reaching 37.04%, so it can be used as fuel to utilize the carbon therein. Although the stone coal is rich in V, the grade of the V resources in different regions is different [32]. The V content in the stone coal of this study sample is 0.45%, which does not reach the industrial grade of V (0.7%) or its cut-off grade (0.5%). Therefore, the stone coal in the study area cannot be directly used as vanadium resource for industrial exploitation. Due to the low content of the Co in the stone coal, its percentage content was not detected by XRF test.

### 3.2. Vanadium and Cobalt in Stone Coal

Bin [33] states that the content of vanadium in China’s stone coal accounts for more than 87% of the world’s total V storage. Stone coal and V titanomagnetite are two important sources of V, and the stone coal mine is a unique new type of V mineral resource in China. It is mainly distributed in stone coal mines in Zhejiang, Anhui, Hunan, Jiangxi, Hubei, Guangxi, Sichuan, Shaanxi and Yunnan provinces. Some of the studies show that the V content in Anhui stone coal can be as high as 8000 μg·g^−1^ [34]. According to the data of the U.S. Geological Survey (USGS), the proven reserves of cobalt resources in the world were about 7.1 million tons in 2015, while the reserves of China’s Co resources are only 80,000 tons, accounting for 1.1% of the world’s total. Moreover, China’s Co resources are mostly from associated minerals of a low grade and low recycling rate [35]. The average contents of V and Co in Chinese coal and worldwide coal are shown in Table 3. The content of V in Chinese coal is significantly higher than that in worldwide coal. The stone coal in this study comes from the open-pit stone coal mine in Haoping river basin, Ankang City, Shaanxi Province in China. The total concentration of V in the raw stone coal sample, before the experiment, was determined to be 1548 μg·g^−1^, and the Co was 3.49 μg·g^−1^ by ICP-MS. By comparison, it can be concluded that the V content in the stone coal of this study was higher than the average content of V in Chinese coal and worldwide coal, which conforms to the rule of a high V content in stone coal. The content of Co in this study was lower than it is in Chinese coal and worldwide coal.

### 3.3. Enrichment of Vanadium in Stone Coal Combustion Products

The relative enrichment factors (EFs) are ordinarily used to evaluate the degrees of the element enrichment during the stone coal combustion, and is defined as:EFs = (C_A_ × A_d_)/C_C_(1)C_C_ and C_A_ are the elemental concentrations in coal and its corresponding combustion ash, respectively. A_d_ is the ash yield of coal [42]. Under normal conditions, 0 < EFs < 1, the large EFs indicate a high enrichment and the low volatility of an element, in other words, the larger the EFs value is, the stronger the enrichment of the element is, and the lower the release is; meanwhile, the smaller the EFs value is, the greater the release is. The ash yield ratio would gradually become large with an increase in the combustion temperature. The ash-based contents of the elements in the heated coal samples need to be converted to the total coal-based content, in order to compare the enrichment characteristics of the element at different heated temperatures. The product of C_A_ and A_d_ was the total coal-based contents of the combusted coal sample. 

The EFs values of V and Co in the stone coal combustion products at each temperature from 500 °C to 1000 °C are shown in Table 4. The EFs of V are greater than 0.79 and that of Co are greater than 0.91 at different combustion temperatures. Overall, the EFs of Co are generally higher than those of V, indicating that V and Co volatilize relatively low during the stone coal combustion. Most of V and Co are rich in combustion products, and the enrichment degree of Co is greater than that of V. The EFs of V decrease slightly and then increase when the temperature rises from 500 °C to 1000 °C; as the temperature rises from 500 °C to 600 °C, the EFs of V rise from 0.79 to 0.87. It is presumed that this phenomenon is caused by the decomposition of low volatile substances. When the temperature rises from 600 °C to 800 °C, the EFs of V decrease from 0.87 to 0.81, and then gradually increase to 0.92 when the temperature was 1000 °C. The EFs of Co reach 0.99 when the temperatures are 500 °C and 600 °C; when the temperatures rise to 700 °C and 800 °C, the EFs decrease slightly to 0.90 and 0.91, respectively; they rise to 0.99 and 0.99 again, when the temperatures are 900 °C and 1000 °C, respectively. According to Liu Chun [42], thermogravimetric analysis (TG) was carried out during the decarbonization process of stone coal analysis roasting, the stone coal will lose weight due to the decomposition of calcite at 756 °C, the thermal stability will be reduced, and the rate of ash reduction will increase, which may be the reason why the EFs of Co first increase and then decrease.

Through the content analysis, the experimental data show that the mass fraction of V in the raw stone coal is 0.45%. The mass fraction of V is 0.62% in the combustion product obtained when the combustion temperature is 600 °C, which has exceeded the 0.5% cut-off grade which can then be used as a mine for V. When the combustion temperature is 1000 °C, the mass fraction of V in the combustion product reaches 1.73%, which is 1.03% higher than the industrial grade of V. Therefore, the combustion products of stone coal have a high industrial value in extracting V.

### 3.4. Chemical Speciation of V and Co in Stone Coal and Its Combustion Products

The sequential chemical extraction method was used to determine the chemical speciation of raw coal and its combustion products obtained at 600 °C and 1000 °C. The presence of V and Co in stone coal and combustion products are shown in Figure 1.

The main speciation of Co in the raw stone coal of this study is residue, accounting for 38.24%; then the organic matter bound state, accounting for 22.42%; Fe-Mn oxide in Co accounts for 18.24%; the carbonate-bound Co accounts for 11.83%; and the exchangeable Co accounts for 8.35%. With the increase in the temperature, the content of the exchangeable Co, organic Co and carbonate-bound Co decreased, and the content of the Fe-Mn oxidized cobalt decreased from 18.24% to 5.96% and then increased to 8.93%, while the content of the residual cobalt increased with the increase in the temperature. When the temperature was 1000 °C, the content of the residual Co accounted for 82.87% of the total Co content.

The dominated speciation of V in raw stone coal of this study is residue, accounting for 88.19%; the second is the organic matter bound state, accounting for 6.43%, similar to a previous study. Dai Zhang [43] has studied the change in the occurrence state of V in stone coal before and after combustion. His research found that V in the raw stone coal mainly exists in the aluminosilicate phase and organic state, some exists in the ferromanganese oxide state and adsorption state, and very little exists in the carbonate state. The organic V content is higher than that in the combustion products and the content of the carbonate-bound V increased with the increase in the combustion temperature, which is presumed to be caused by the decomposition of organic matter during combustion, and the decomposition of clay minerals.

### 3.5. Mineral Compositions of Stone Coal and Its Combustion Products

In order to further explore the behaviors of V and Co enrichment in stone coal during combustion, the mineral compositions and micro structures were characterized by XRD and SEM analysis. The SEM photomicrographs of RC, P600 and P1000 are shown in Figure 2.

The analysis of the raw stone coal and its combustion products showed that the V content in RC, P600 and P1000 were 0.45%, 0.62% and 1.73%, respectively. The content of Co has not been analyzed, so the phase composition of V and Co cannot be measured when mineral composition analysis is performed. According to XRD detection analysis (Figure 3), the main minerals in the raw stone coal are quartz, clay minerals (kaolinite, illite), and oxybiotite. Hematite was detected in the combustion products of 600 °C and 1000 °C except the above minerals, and the content increased with the increase in the temperature. Quartz is the main mineral phase, which was consistently present in the raw stone coal and combustion products. With the increase in the temperature, illite was initially dehydrated at a high temperature and then formed a glass phase at 950 °C, followed by the final generation of silicon spinel, hematite, quartz and other minerals [44]. The hematite was formed during the process of mineral restructure and reform when the stone coal was combusted at a high temperature [45,46]. At high temperature, biotite will decompose, and the fluorine and hydroxyl groups in biotite may escape as gaseous fluoride and water vapor [43].

It is reported that the V in lignite generally has an organic affinity, while in bituminous coals and anthracites, it is associated with clays [47]. Most scholars have completed research for the occurrence state of cobalt in coal [41,45,46].

It is generally believed that the Co in bituminous coal and anthracite mainly exists in the form of sulfide and arsenide, followed by aluminosilicate and the organic state [46,48]. Stone coal is a kind of inferior anthracite. The spectrogram clearly shows V, Si, K, Fe, Mg, Al and other elements, from which it can be clearly seen that V is often accompanied by other elements, such as Si and Al, which indicates that a large part of the V in stone coal and its combustion products is located in the aluminosilicate phase. This explains the results of the continuous chemical extraction experiments and is consistent with previous studies [46,49,50,51].

### 3.6. Potential Impact of Stone Coal Combustion on Surrounding Environment

Some studies have shown that the residual state is almost insoluble in aqueous solution, and its bioavailability is low, making it difficult to migrate in the environment. Exchangeable state is soluble in aqueous solution, its bioavailability is the highest among the several chemical forms and is easy to migrate. The solubility and bioavailability of the organic matter-bound state, Fe-Mn oxide-bound state and carbonate-bound state are between the above two, and can partially migrate in the environment [52,53]. It can be seen that the content of V in stone coal and its combustion products largely exceeds the local soil background value; the bioavailable V content in the combustion products obtained from stone coal at 1000 °C in this study area is 2.73%, which is higher than that of the raw stone coal. The V released from mining activities can enter into the food chain through a series of physical and chemical changes after entering the soil, thus it might endanger human health. The bioavailable Co in the raw coal is higher than that in combustion products. Therefore, if the stone coal and its combustion products in this study area are stored without effective treatment, V and Co will migrate into the surrounding soil and groundwater environment, through leaching and other methods, to reduce the quality of the surrounding environment. The plants growing around will absorb the V and Co in the soil and groundwater, and the V and Co in plants will enter the human body through the food chain, then impact on human health. The potential environmental risk of vanadium is higher than that of cobalt.

## 4. Conclusions

This study clarified the enrichment and chemical speciation of V and Co in stone coal and its combustion products in southern Shaanxi. The results show that the content of V in raw stone coal is 1548 μg/g, which is higher than its average content in Chinese coal and worldwide coal, but it does not reach its industrial grade. The content of Co in raw stone coal is 3.49 μg/g, lower than its average content in Chinese coal and worldwide coal. The enrichment factors (EFs) of V in combustion products at different temperatures are greater than 0.79, and the level reached 0.92 at 1000 °C. The EFs of Co are generally higher than V, greater than 0.91 and the highest can be 0.99, indicating that V and Co volatilization is relatively low during stone coal combustion. Most of the V and Co are rich in combustion products and the enrichment capacity of Co is higher than V. It is worth noting that the content of V in 600 °C combustion products reaches its cut-off grade, and the content of V in 1000 °C combustion products exceeds its industrial grade. Therefore, the stone coal in this region can not only be used as an energy mineral, but also combustion products are valuable in extracting and utilizing V. Five operationally defined modes of the chemical forms of V and Co were extracted from the stone coal and its combustion products, using the sequential chemical extraction method. Among the five speciations (exchangeable state, carbonate binding state, Fe-Mn Oxides and residue state), the main chemical speciation of V is the residue state. The content of the organic V decreases and the content of the carbonate state V increases with the increase in the combustion temperature. The main speciation of Co in raw stone coal is the residue state, followed by the organic matter bound state. With the increase in the temperature, the content of the exchangeable Co, organic Co and carbonate-bound Co decreases, and the content of Fe-Mn oxidized Co decreases firstly and then increases, while the content of the residual Co increases with the increase in the temperature. It is worth noting that the content of V in 600 °C combustion products reaches its cut-off grade, and the content of V in 1000 °C combustion products exceeds its industrial grade. At the same time, it should be noted that, if the combustion products are stored without effective treatment, the surrounding environment will be polluted and human health will be endangered.

## Figures and Tables

**Figure 1 ijerph-19-09161-f001:**
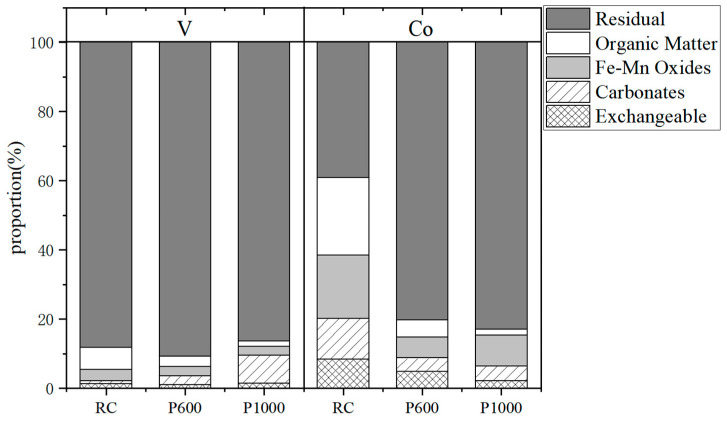
Speciation fractions of V and Co in stone coal (RC) and its combustion products at 600 °C (P600) and 1000 °C (P1000).

**Figure 2 ijerph-19-09161-f002:**
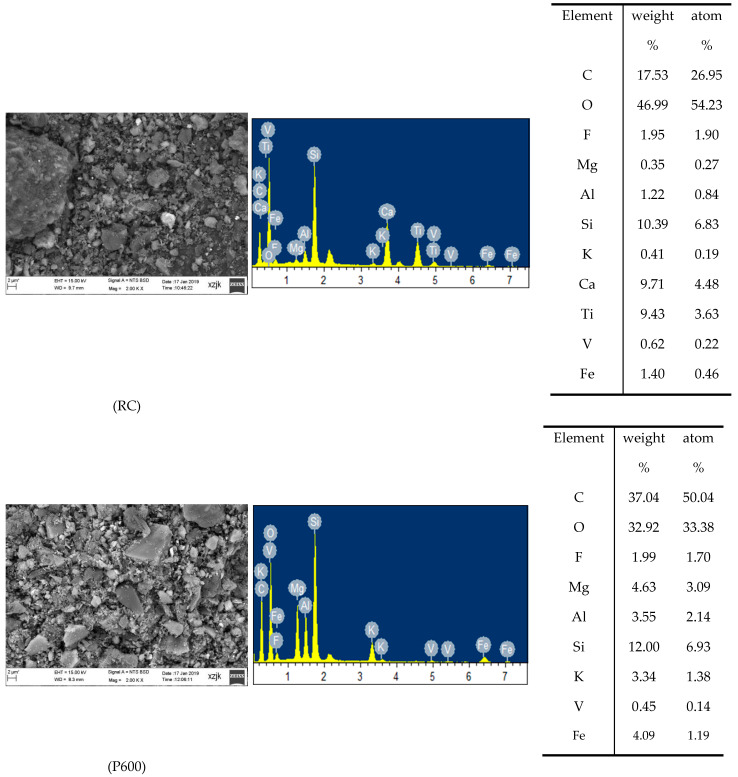
SEM photomicrograph of RC, C6 and C10 produced in the laboratory (tables on the right show the composition of SEM energy spectrum point scanning results, including weight percentage and atomic percentage).

**Figure 3 ijerph-19-09161-f003:**
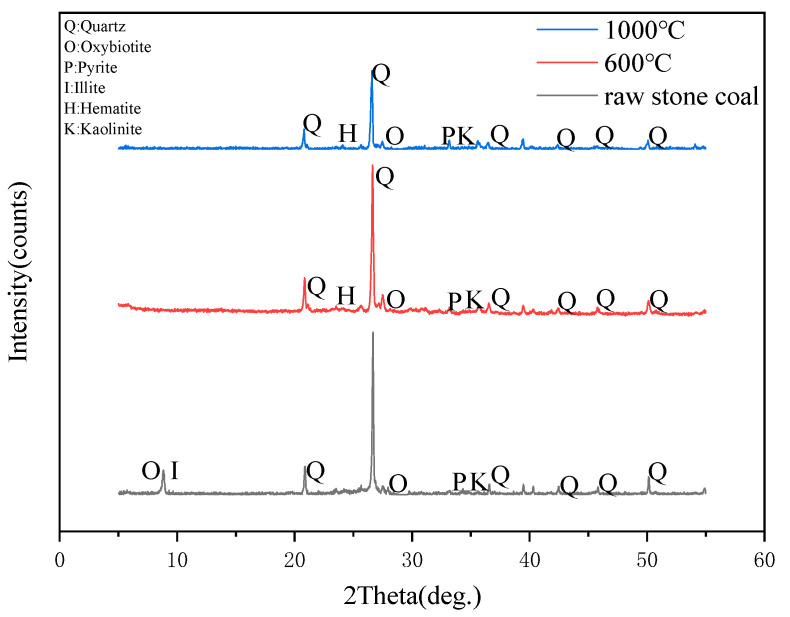
XRD patterns of raw stone coal, combustion residue after burning at 600 °C and 1000 °C.

**Table 1 ijerph-19-09161-t001:** Proximate and ultimate analysis in the raw coal sample.

M_ad_ (wt. %)	A_d_ (%)	V_daf_ (wt. %)	F_Cd_ (wt. %)	S_t,d_ (%)	S_s,d_ (wt. %)	S_p,d_ (wt. %)	S_o,d_ (wt. %)	Q_gr,ad_ (MJ/kg)
2.13	38.96	4.66	51.41	1.82	0.03	1.76	0.03	5.53

Among which, ad stands for air dry basis; d stands for dry; daf stands for dry ash free; M stands for moisture; A stands for ash yield; V stands for volatile matter; F_C_ stands for fixed carbon; S_t,d_ stands for total sulfur; S_p,d_ stands for pyritic sulfur; S_s,d_ stands for sulfate sulfur; S_o,d_ stands for organic sulfur; Q_gr,ad_ stands for high calorific value of air drying base.

**Table 2 ijerph-19-09161-t002:** Chemical analysis of the raw stone coal.

Element	C	O	Fe	F	Mg	Al	Si	K	V	Co
ω (%)	37.04	32.92	4.09	1.99	4.63	3.55	12.00	3.34	0.45	ND

ω stands for Mass percentage; ND stands for not detected.

**Table 3 ijerph-19-09161-t003:** Average contents of Co and V in Chinese coal and worldwide coal.

	V (μg·g^−1^)	Co (μg·g^−1^)	References
Stone coal	1548	3.49	In this study
World coal	25.0	5.1	Ketris and Yudovich, 2009 [36]
Chinese coal	51.81	7.08	Dai et al., 2012 [37]
			Liu Yuan, 2018 [38]
US coal	22	6.10	Finkelman, 1993 [39]
Australian coal	25	4.00	Raask, 1985 [40]
			Swain, 1990 [41]

**Table 4 ijerph-19-09161-t004:** Contents and enrichment factors (EFs) of V and Co in raw coal and combustion products.

Samples	Ad/%	C_v_/μg·g^−1^	EFs (V)	C_Co_/μg·g^−1^	EFs (Co)
RC	39.16	1548	-	3.49	°C
P500	69.52	1771	0.79	5.02	0.99
P600	61.83	2183	0.87	5.57	0.98
P700	57.46	2309	0.86	5.48	0.90
P800	48.55	2589	0.81	6.55	0.91
P900	40.02	3188	0.82	8.72	0.99
P1000	38.21	3739	0.92	9.03	0.99

## Data Availability

Data available from corresponding authors.

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
