# Peer review of "Enrichment and Chemical Speciation of Vanadium and Cobalt in Stone Coal Combustion Products in Ankang, Shanxi Province, China"

_ijerph, 2022, doi:10.3390/ijerph19159161_

Round 1
Reviewer 1 Report
Simulation combustion experiments and sequential chemical extraction were conducted in this manuscript. The enrichment, chemical form, and form-related environmental risks of rough coal and its combustion products at different combustion temperatures were studied. The study contributes to understanding the industrial value of stone coal combustion products and possible environmental risks.
(1) Provide some detailed info about the effects of geological ages or regions on coal combustion in the Introduction Section. On this basis, please consider the papers presented below.
Impact of the crystallite parameters and coal ranks on oxidation and combustion properties of Carboniferous coals and Jurassic coals. Arab J Geosci 11, 662 (2018).
Comparison of combustion characteristics and kinetics of Jurassic and Carboniferous-Permian coals in China,Energy,2022,124315
(2) In the simulated combustion experiment in the manuscript, 6 combustion temperature points were selected for the experimental test. Why are the research results only analytically discussed at 500, 600, and 1000°C?
(3) How to conclude that the sulfur in the raw stone coal is mainly pyrite sulfur after proximate analysis?
(4) “The EFs of Co reaches at 0.99 when the temperature is 500℃ and 600℃, when the temperature rises to 700℃ and 800℃, it decreases slightly to 0.96 and 0.91, respectively; it rises to 0.99 and 0.98 again when temperature is 900℃ and 1000℃, respectively.” Please give a reasonable explanation.
(5) The experimental data of XRD in the manuscript is not displayed in the form of tables or figures It is unclear to the readers.
(6) “The content of Fe-Mn oxidized cobalt decreased from 18.24% to 5.96% and then increased to 8.93%”, which doesn’t match the data in Figure 1. Please check carefully.
(7) Line 224: 100°C should be revised to 1000°C.
Author Response
Thank you for your affirmation of our manuscript entitled "Enrichment and chemical speciation of Vanadium and Cobalt in stone coal combustion products in Ankang, Shanxi Province, China". Thank you for your comments,which are of great significance to the improvement of the quality of our paper. We have studied these comments carefully and have made correction which we hope meet with approval. Revised portion are marked in red in the revisions. In addition, we have thoroughly checkd the grammar error and ask a native English speaker to polish the Language.Please check the Word document for details.

Reviewer 2 Report
The manuscript is about enrichment and chemical speciation of Vanadium and Cobalt in stone coal combustion products in China. The scope of this article is consistent with the requirements of the International Journal of Environmental Research and Public Health, but it requires major revision in accordance with the comments below:
11. Avoid linking references as in: [1-3], [4-6], [38-40]. Please combine no more than two references. Instead summarise the main contribution of each referenced paper in a separate sentence.
2 2. Section 2 should be Materials and Methods.
33. Please add the references to all coal standards described in point 2.2.
44. The abbreviations:
- sodium acetate should be NaOAc, not NaAc,
- acetic acid should be HOAc, not HAc,
- ammonium acetate should be NH4OAc, not NH4Ac.
5. Table 1 - should be subscripts.
6. Please standardize the units in line 212.
7. Table 3 – should be US coal, not US cola.
8. Figure 1 – I think it should be here proportion, not proporation.
9. I think it would be a good idea to find and cite two articles from IJERPH in a similar scientific area.
Author Response

(The authors gave the same response as above.)

Reviewer 3 Report
Comments to the manuscript “Enrichment and chemical speciation of Vanadium and Cobalt in stone coal combustion products in Ankang, Shanxi Province, China” submitted by Wei Cui, Qingjun Meng, Wenbo Li and Qiyan Feng
Stone coal is a very interested raw material containing high concentration of metals. The manuscript contains interesting data but it is not suitable for publication without serious reworking and additional analyses.
General comments:
1. I suggest to add more data on the “stone coal” – rock characteristics, occurrence composition, etc.
2. Results are based on very limited number of samples. In line 84: “Six replicates’ samples of raw stone coal were collected”. One sample was divided into six subsamples to compare results? Number of results presented in the manuscript indicates that only one sample was analysed. Sampling of the outcrop (mine pit) is of crucial importance for interpretation. To obtain representative results it is necessary to prepare suitable sampling strategy. Bigger number of samples is needed for analysis and results discussion.
3. Chemical data are unclear. In Table 2 concentration of V in the studied sample is equal to 0.45% (weight %?). In lines 212 the value for V is 1548 mg/g-1 and Co 3.49 mg/g-1 (for analyzed sample). In Table 3 values 1548 mg/g-1 (for V) and 3.49 mg/g-1 (for Co) are cited as data from Ketris and Yudovich 2009. It is necessary to explain it.
4. Method of the calculation of enrichment factor. The formula indicated in the line 223 can be applied but comparison of relative values can give better results. Celement/CAl in ash compared with Celement/CAl in raw coal. Precision of the determination of the ash content is usually limited.
5. Results of XRD analysis must be rewritten. “The XRD analysis of the raw stone coal and its combustion products showed that the V content in RC, P600 and P1000 were 0.45%, 0.62% and 1.73% respectively.” (line 291). XRD is not a method of chemical analysis. Mineral composition is rather simply. Please define criteria of differentiation of kaolinite and dickite. It is commonly accepted to present a X-ray diffractogram in the manuscript. I suggest to follow this “tradition”.
“illite was initially dehydrated at a high temperature and then forms a glass phase at 950°C followed by the final generation of silicon spinel, hematie, quartz and other minerals” (lines 300 and 301). Conclusions are not supported by results. Can you prove the presence of glass or spinel?
“Al of plagioclase might be replaced by Fe and the hematite was formed during the process of mineral restructure and reform when the stone coal was combusted at a high temperature” (lines 302 and 303). Al in plagioclase is replaced by Fe? Can you find evidences of this process? Do you suggest relationship between hematite crystallization and plagioclase decomposition?
6. SEM-EDS analyses are used to support an idea that “large part of V in stone coal” is located in aluminosilicate phase. Three EDS spectra (Figure 2) are included. Two of them (RC and P600) represent material rich in Si, with minor content of Al and other elements. The third spectrum represent material rich in Ca and S, probably calcium sulphate (and aluminosilicate phase is a minor component). Generally it is difficult to interpret spectra. It is strongly recommended to present results of chemical analysis in the table with contents of various elements. There is not information about the area analyzed (spot analysis; large area?). SEM images represent strongly different magnifications. Spectra are probably related to selected grains (small rectangles on SEM images). Results do not support the conclusion that V occurs mainly in aluminosilicates (but it cannot be ruled out).
7. Linguistic correction of the manuscript is needed.
Detail comments:
Line 24. “ancient strata” – term is not in use for determination of the age of geological strata
Line 120. “order to study the five chemical speciation of V and Co”. I suggest to change to “concentration of V and Co in five solutions obtained in sequential extraction and representing five fractions”
Line 158: “Known” instead of “know”
Line 184: “The V content in stone coal of this study region is 0.45%” or “in the studied sample”
Line 187: “Due to the low content of Co, its percentage content was not analyzed in the chemical element analysis”. It wasn’t analyzed because low content was expected? It was analyzed but the content was below detection limit? Please give the detection limit value.
Line 194 and 195: It is not necessary to explain the meaning of symbols of chemical elements.
Table 3. “Ketris and Yuclovich,2009” should be “Ketris and Yudovich, 2009”; “Word coal” – perhaps “World coal”; “US cola” – perhaps “US coal”
Author Response

(The authors gave the same response as above.)

Round 2
Reviewer 2 Report
The manuscript has been sufficiently improved and can be publicate in the present form.
Author Response
Thank you for your approval of the revised manuscript.The improvement of the article cannot be separated from your valuable suggestions.We corrected some reference formatting problems and checked the spelling of the text again.
We all authors extend our sincere thanks to you.